# Whole-Genome Sequencing of Two Canine Herpesvirus 1 (CaHV-1) Isolates and Clinicopathological Outcomes of Infection in French Bulldog Puppies

**DOI:** 10.3390/v16020209

**Published:** 2024-01-30

**Authors:** Angela Maria Rocchigiani, Loris Bertoldi, Elisabetta Coradduzza, Giada Lostia, Davide Pintus, Rosario Scivoli, Maria Giovanna Cancedda, Mariangela Stefania Fiori, Roberto Bechere, Anna Pina Murtino, Giovanni Pala, Giusy Cardeti, Simona Macioccu, Maria Antonietta Dettori, Antonio Pintore, Ciriaco Ligios, Giantonella Puggioni

**Affiliations:** 1Istituto Zooprofilattico Sperimentale della Sardegna, 07100 Sassari, Italy; angelamaria.rocchigiani@izs-sardegna.it (A.M.R.); giada.lostia@izs-sardegna.it (G.L.); davide.pintus@izs-sardegna.it (D.P.); rosarioscivoli@gmail.com (R.S.); mariagiovanna.cancedda@izs-sardegna.it (M.G.C.); mariangela.fiori@izs-sardegna.it (M.S.F.); roberto.bechere@izs-sardegna.it (R.B.); annapina.murtino@izs-sardegna.it (A.P.M.); giovanni.pala@izs-sardegna.it (G.P.); simona.macioccu@izs-sardegna.it (S.M.); mariaantonietta.dettori@izs-sardegna.it (M.A.D.); antonio.pintore@izs-sardegna.it (A.P.); ciriaco.ligios@izs-sardegna.it (C.L.); giantonella.puggioni@izs-sardegna.it (G.P.); 2BMR Genomics s.r.l., 35131 Padova, Italy; loris.bertoldi@bmr-genomics.it; 3Istituto Zooprofilattico Sperimentale del Lazio e della Toscana M. Aleandri, 00178 Roma, Italy; giusy.cadeti@izslt.it

**Keywords:** canine herpesvirus, phylogeny, variants analysis

## Abstract

*Canine herpesvirus 1* (CaHV-1) infects dogs, causing neonatal death and ocular, neurological, respiratory, and reproductive problems in adults. Although CaHV-1 is widespread in canine populations, only four studies have focused on the CaHV-1 whole genome. In such context, two CaHV-1 strains from both the kidney and spleen of 20-day-old deceased French Bulldog puppies were recently isolated in Sardinia, Italy. The extracted viral DNA underwent whole-genome sequencing using the Illumina MiSeq platform. The Italian CaHV-1 genomes were nearly identical (>99%), shared the same tree branch, and clustered near the ELAL-1 (MW353125) and BTU-1 (KX828242) strains, enlarging the completely separated clade discussed by Lewin et al., in 2020. This study aims to provide new insights on the evolution of the CaHV-1, based on high-resolution whole-genome phylogenetic analysis, and on its clinicopathological characterization during a fatal outbreak in puppies.

## 1. Introduction

Canine herpesvirus 1 (CaHV-1) is a *Varicellovirus* of the *Orthoherpesviridae* family (also called *Varicellovirus Canidalpha1* according to the latest revision of ICTV), that affects neonatal and adult dogs [1]. The virus is responsible for sudden death in puppies, characterized by the diffusion of progressive multifocal hemorrhagic necrosis of multiple organs [2,3], and for the onset of ocular [4], respiratory [5,6], and reproductive disorders in adults [1,7,8]. CaHV-1 infection can be confirmed with a serum neutralization test (SNT), enzyme linked immunosorbent assay (ELISA), virus isolation, fluorescent antibody techniques, electron microscopy, real-time PCR assay, and polymerase chain reaction (PCR is the most reliable method in adults [3]). There is a vaccine available which, however, does not ensure total protection, so at the moment the prevention of this contagion is the only useful way to reduce the spread of the virus. Like all Herpesviruses, once infected, the canine herpesvirus remains latent in the animal’s body and the viraemia can exacerbate during the presence of stress or of a weakened immune system [9]. The CaHV-1 genome is estimated to be approximately 125 kbp with 76 open reading frames [1]. The genome is composed of a unique long sequence (UL), of about 97.5 kbp and a unique short sequence (US), of about 7.7 kbp, which are flanked by 38 kbp and 10 kbp terminal (TRS/TRL) and internal inverted repeats (IRS/IRL), respectively [1,10]. Even though CaHV-1 is widespread in canine populations, only a small number of studies have been published [1,2]. Phylogenetic inferences based on whole genomes have been found in four of these studies, which include a total of 20 novel CaHV-1 isolated genomes: three from UK in 2016 (CaHV-1/194, CaHV-1/V777, CaHV-1/V1154) [1]; one from Brazil in 2016 (BTU-1) [11]; one from New Zealand in 2018 (CaHV-1/15-4016-NSW) [12]; and fifteen from USA in 2020 (ELAL 1-15) [1].

In this study, two CaHV-1 whole genomes were isolated from the kidney and spleen of 20-day-old French Bulldog puppies. Isolated sequences were compared with all of the CaHV-1 [1] genomes currently available in the public database to perform a complete viral phylogenetic analysis. In addition, the application of whole-genome sequencing provided many specific details, especially after translation of genes encoding for proteins such as large integument protein (pUL36), DNA tegument packing protein (pUL25), envelope glycoprotein H, helicase-primase subunit (pUL8) and virion (pUS2), comparing them with homologous ones belonging to human herpesviruses [13,14,15,16,17,18], which have been studied for longer. Evidence of nonsynonymous amino acid substitutions or amino acid insertions (as in pUS2), although at positions not included in catalytic or otherwise functional sites, characterizes our strains compared with those in the bibliography [1]. The data, albeit from only two genomes, are distinctive features of our sequenced strains.

Herein, we provide new insights on the evolution of the CaHV-1, based on high-resolution phylogenetic analysis of whole genomes, and the characterization of its clinicopathological presentation during a fatal outbreak in puppies.

## 2. Materials and Methods

### 2.1. Case and Sampling

A French Bulldog litter from an amateur breeding farm in Sardinia presented a case of clinical syndrome that resulted in a fatal exitus. The French Bulldog puppies were not feeding and showed acute respiratory symptoms, pain, and yellow foamy diarrhea. The farm consisted of 12 healthy animals, 2 males and 10 females ranging from 4 months to 4 years of age, and a 5-puppy litter showing a clinical syndrome. The mother of the newborns was a two-year-old specimen, and the fertilization mode used was artificial fertilization. A total of three of the affected puppies were submitted to necroscopy soon after their death.

Necropsy sampling on puppies included liver, kidneys, lungs, brain, thymus, heart, retropharyngeal lymph nodes, and third eyelid and chest cavity swabs. On the breeding subjects on the other hand, serum and blood collection and vaginal, preputial, and ocular swabs were performed. In addition, vaginal, tracheal, and ocular swabs from the mother were taken for viral isolation and CaHV-1 identification through real-time PCR. Most of the collected samples were stored at −80 °C.

### 2.2. Histopathology

Tissue samples for histological examination were taken from the abovementioned organs and fixed in 10% formalin, included in paraffin by routine methods, then cut into 4 μm thick sections and stained using the ST Infinity Hematoxylin and Eosin Staining System (Leica Biosystem, Richmond, IL, USA). Stained sections were observed under a light microscope in a magnification range of 50× to 400×.

### 2.3. RNA In Situ Hybridization

The detection of CaHV-1 was performed using the RNAscope 2.5 Red assay kit (Advanced Cell Diagnostics [ACD] Cat No. 322360, Hayward, CA, USA). The RNAscope probe was designed by Advanced Cell Diagnostics to detect the mRNA expression of the Canine herpesvirus thymidine kinase gene (V-CHV-TK) (Cat No. 457121). Deparaffinized and dehydrated 5 µm thick tissue sections were incubated in hydrogen peroxide for 10 min and soon after in a target retrieval buffer (ACD) at 98 °C/100 °C using a hot plate for 15 min. Tissue sections were then treated with a Protease Plus reagent (ACD) in a hybridization oven at 40 °C for 25 min (ACD, Hayward, CA, USA) and subsequently incubated at 40 °C with the target probe in the same hybridization oven for 2 h. A probe targeting the PPIB gene (Cyclophilin B) and a probe targeting the DapB gene (*Bacillus subtilis* strain SMY) were used as positive and negative controls, respectively. A series of target-specific signal-amplification steps were conducted for both assay kits and Fast Red signal-amplification system was then hybridized to the target probes. Slides were counterstained with hematoxylin and cover slipped using Ecomount (Biocare Medical, Concord, CA, USA). The sections were examined under a light microscope (Leica DM4000B).

### 2.4. Virus Isolation

CaHV-1 was isolated on MDCK cells from a tissue homogenate of brain, lung, thymus, kidney, heart, and spleen, according to the methodology reported by Kawakami et al., 2010 [19]. Briefly, a total of 0.5 g of each tissue sample was homogenized in 5 mL (10% *w*/*v*) of Phosphate Buffered Saline (PBS) medium supplemented with the following antibiotics: 400 UI/mL penicillin, 400 g/mL streptomycin, 300 g/mL gentamicin, and 2.5 g/mL amphotericin B. The suspension obtained was centrifuged at 1000× *g* for 15 min and used for the infection of the MDCK cells using cells from the 100th to the 120th passage. The MDCK cells were cultured and maintained in an MEM cell culture medium (Corning) supplemented with 10% fetal bovine serum (FBS) (GIBCO). Semiconfluent MDCK cells on 25-cm^2^ flasks were washed with PBS and inoculated with 500 μL of sample, incubated for 1 h in a thermostat at 37 °C and 5% CO_2_. A volume of 7 mL of fresh MEM with 1% (*v*/*v*) FBS and antibiotics (final concentration of 100 UI/mL penicillin, 100 g/mL streptomycin, and 0.5 g/mL amphotericin B) were added and incubated under the same conditions. Furthermore, 500 μL of PBS solution was used as negative control and 500 μL of CaHV-1 23612/24432 strain was used as positive control. A cytopathic effect (CPE) was observed daily until the 5th day of culture. When no CPE was evident, a blind passage was made. As soon as CPE was detected, all flasks were freeze-thawed three times, and clarified by centrifugation at 800× *g* for 10 min at room temperature. The collected supernatants from flasks with CPE were stored at −80 °C until virus identification by real-time PCR.

### 2.5. Real Time PCR

Nucleic acid purification from samples was performed using the MagMax Core Nucleic Acid Purification Kit (Applied Biosystems-ThermoFisher Scientific, Waltham, MA, USA) in an automated sample preparation workstation MagMAX Express 96 (Applied Biosystems), according to the manufacturer’s instructions. The absolute real-time PCR protocol was carried out using primers and probes published in Decaro et al., 2010 [20] and performed using a 7900 HT Fast Real-Time PCR System (Applied Biosystems, Waltham, MA, USA). A positive CaHV-1 result was considered when Ct values were ≤35. The absolute real-time PCR protocol was carried out using primers and probes published in Decaro et al., 2010 [20] and performed using a 7900 HT Fast Real-Time PCR System (Applied Biosystems, Waltham, MA, USA). A positive CaHV-1 result was considered when Ct values were ≤35.

SsoAdvanced^TM^ Universal Probes Supermix (BioRad Laboratories, Hercules, CA, USA) were used for this assay. For a differential diagnosis, other pathogens like Leptospira and Parvovirus were investigated, which were negative, as was the culture examination.

### 2.6. Serum Virus Neutralization Test

A total of 10 female and 2 male healthy dogs were serum-sampled to evaluate the response to CaHV-1 through the serum neutralization test, using a modified version of the methodology from Rota A. et al., 2020 [21]. In brief, the sera were inactivated for 30 min in a water bath at 56 °C. The two-fold serum dilutions in 50 µL MEM cell culture medium (Corning) from 1/4 to 1/1024 were performed in two rows in 96-well tissue culture microplates (Greiner). Positive and negative control serum, cell control, and viral dilution control were included in each test. An extra well with undiluted test serum was used as the serum toxicity control. Subsequently, 50 μL of 100 TCID50 of the CaHV-1 23612/24432 strain were added to each well, except for the ones containing the serum toxicity control and the cell control. After a 1 h incubation at 5% CO_2_ and 37 °C, 100 μL of an MDCK cell suspension at a concentration of 2 × 10^5^ cells/mL in MEM + 10% fetal bovine serum (Gibco) was added to each well containing the virus–serum mixture. Microplates were then placed in a CO_2_ incubator at 37 °C and microscopic examination for the presence of a characteristic CaHV-1 CPE was performed after 3, 4, and 5 days of incubation. The SN titer was defined as the reciprocal of the highest serum dilution inhibiting CPE in 100% of the cell cultures.

Titers equal to or greater than 1:4 were considered positive for exposure to CaHV1. Antibody titers were categorized in the following classes: 1:4–1:8 weakly positive; 1:64–1:128 positive; 1:256 strongly positive.

### 2.7. Sequencing, Genome Assembly and Phylogeny

Sequencing was performed at BMR Genomics s.r.l. (Padova, Italy). DNA from the spleen and kidney of the two puppies were extracted and purified. The CELERO PCR Workflow with enzymatic fragmentation was used to prepare two Illumina libraries, following the manufacturer’s instructions. Subsequently, the libraries were checked with a BioAnalyzer (Agilent, Santa Clara, CA, USA) and quantified with the Qubit fluorometer (Thermo Fisher, Bedford, MA, USA). Finally, they were pooled with other samples, loaded into the Illumina MiSeq sequencer (Illumina, Inc., Ann Arbor, MI, USA), and sequenced according to the V3-300PE strategy.

Reads quality was firstly assessed with FastQC (v0.11.9) [22], followed by the preprocessing step performed with fastp (v0.23.2) [23], where residual primer sequences, low-quality bases (Q < 30), and short reads (length < 150 bp) were removed.

Furthermore, reads were mapped with bowtie2 (v2.4.5) [24] using standard parameters on the reference genome of *Canis lupus familiaris* (Canis_lupus_familiaris.ROS_Cfam_1.0.dna.toplevel.fa) to remove the host contaminants.

Finally, the properly unmapped reads (samtools view-f 13) [25] were collected and aligned again with bowtie2 against the reference genome of the CaHV-1 strain 0194 (GCF_001646155.1/NC_030117.1). After this step, properly unmapped reads were removed (samtools view-f 3-F 12) so that only the reads that were at least once mapped to the reference were kept. This procedure was also tested, using as reference all the CaHV-1 available assemblies, without showing significant differences.

Cleaned reads of both samples underwent two parallel steps of de novo assembly, with SPAdes (v3.14.5, with careful option) [26], without using a reference genome, and with Geneious Prime (2022.1.1), exploiting the aforementioned CaHV-1 genome as a guide. Assembly metrics were calculated with QUAST (v5.0.2) [27] and genome completeness was estimated using Busco (v5.3.0) [28] against the herpesviridae_odb10 lineage. Furthermore, gene prediction and annotation were performed using prokka (1.14.6) [29] (kingdom Virus–gcode 1) and eggNOG-mapper (v2.1.7) [30], respectively.

A multiple sequence alignment including all the 20 CaHV-1 genomes reported by Lewin and colleagues [1], the 2 reference-guided assembled sequences of the Sardinian strains, and the Feline herpesvirus (FHV-1) reference sequence (GCF_000885455) used as outgroup, was performed using MAFFT (v7.505) and further processed with MEGAX (v11.0.13) [31] in order to perform an evolutionary analysis. The evolutionary history was inferred using the maximum likelihood method and general time-reversible model [32]. Initial tree(s) for the heuristic search were obtained automatically by applying neighbor join and BioNJ algorithms to a matrix of pairwise distances estimated using the maximum composite likelihood (MCL) approach, and then selecting the topology with superior log likelihood value among the 1000 bootstrap replicates. A discrete gamma distribution was used to model evolutionary rate differences among sites (5 categories (+*G*, parameter = 1.7718)). All of the positions with less than 95% site coverage were eliminated (partial deletion of gaps).

Cleaned reads were also used to further analyze samples at the phylogenetic level. Snippy-multi (v4.0.2) [33] was applied with standard parameters on the reads of the two sequenced strains, along with all the nucleotide sequences of the 20 CaHV-1 genomes. The multiple sequence alignment file was subsequently cleaned with a snippy utility (snippy-clean_full_aln) and used as input for the FastTree software (v2.1.11) [34] in order to build a phylogenetic tree applying the generalized time-reversible model (-gtr). The tree was drawn and refined with the iTOL web tool [35]. The results of the snippy-multi were also used to perform a first step of variant analysis, exploiting custom Python scripts to further process such data. In addition, a second step of variant analysis, focusing mainly on coding regions, was performed through individual alignments of amino acid sequences, translated from each coding DNA sequence of our strains, with protein sequences of all the CaHV-1 strains available on GeneBank. The locations of each open reading frame (ORF) encoding functional proteins were identified by assigning all ATG-initiated ORFs larger than 50 codons for similarity with the reference sequence. Analysis was performed using BioEdit (version 7.0.4.1) and the EMBL-EBI nucleotide sequence translation tool, EMBOSS transeq (http://wwwdev.ebi.ac.uk/Tools/jdispatcher/st/emboss_transeq (accessed on 25 February 2023).

## 3. Results

### 3.1. Clinical and Anatomo-Histopathological Findings, and RNA In Situ Localization

External post-mortem examination of the affected puppies showed mild serous ocular and nasal discharge (Figure 1A), underlining respiratory distress, and swelling in the abdomen, suggestive of ascites and abdominal pain. All of the puppies showed sero-hemorrhagic fluid in the abdominal and thoracic cavity (Figure 1B), hepatomegaly, splenomegaly, and bilateral renomegaly with diffused petechial and ecchymotic hemorrhages in the cortex (Figure 1C).

Histologically, multifocal perivascular necrosis and hemorrhages with scarce mononuclear cell infiltration were identified in several organs, mainly in the kidneys, lungs, and liver (Figure 2A–D). Eosinophilic intranuclear inclusions were detected in hepatocytes adjacent to the necrotic areas (Figure 2E). Moreover, in the lymphatic organs, we observed severe thymic medullary atrophy and depletion of white pulp along with focal areas of necrosis in the spleen (Figure 2F,G). A mild and diffused non-suppurative meningoencephalitis with glial nodules was observed in the brain (Figure 2H). Finally, interstitial edema in the myocardium was detected (Figure 2I).

Significant ISH staining was found in all three of the tested puppies. CaHV-1 viral mRNA was detected proximal to perivascular necrosis and exclusively in hepatocytes and renal tubular epithelial cells. Moreover, inflammatory cells and endothelial cells tested positive for CaHV-1 in the lung, kidney, and liver (Figure 2J–L).

### 3.2. Virus Isolation

The CaHV-1 was isolated from the MDCK cells in 18 out of the 20 examined samples. A cytopathic effect was detected in the confluent monolayer, approximately two days after infection, as rounding, ballooning, grape-like clusters gathered around a hole in the monolayer (Figure 3).

### 3.3. Virological Analyses

Samples from the entire set of puppy organs (brain, lung, liver, thymus, kidney, heart, spleen; Figure 4), as well as the mother’s third eyelid were positive for CaHV-1 using real-time PCR detection. Interestingly, all blood and swab samples from the rest of the breeding members were CaHV-1 negative.

### 3.4. Serological Analyses

With the exception of one four-year-old female, the serum virus neutralization test was positive for all breeding members—including the mother of the puppies.

### 3.5. Reads Preprocessing

Whole-genome sequencing (WGS) of the two CaHV-1 strains under analysis (referred to identifiers OP997219 and OP997220) yielded 1,068,925 and 976,455 300 bp paired-end (PE) Illumina reads, respectively. The preprocessing steps removed a great number of reads (Appendix A) from both samples, keeping only 108,143 and 258,452 PE reads.

Most of the discarded readings can be associated with host contamination, highlighting the still-present difficulties to highly purify viral particles. In addition, a particular consideration must be conducted on the preprocessing step where reads were aligned to the reference genome. In fact, although such procedures could trigger the possible loss of some interesting new regions, it clearly improved the overall assembly quality, leading to the decreasing misassembled contigs.

### 3.6. Genome Assembly

The cleaned reads were assembled into contigs by independently using SPAdes and Geneious Prime. In the latter case, the CaHV-1 reference genome (NC_030117) was used to guide the assembly process. Relevant results for both samples were obtained with the use of both tools (L50 = 1, 100% Busco completeness), even if the use of a genome scaffold clearly improved the result. In fact, contrary to Geneious, which produced a sequence of 125 kbp length, SPAdes was not able to disentangle the inverted terminal repeats at the 3′ end of the viral nucleotide sequence (Appendix A [36]), resulting in a genome of 114 kb. The real viral genome structure was not able to be completely solved by this method, not even after a further scaffolding step with RagTag [37].

Information on the two assembled genomes (samples OP997219 and OP997220) using Geneious Prime is collected in Table 1, which also includes gene predictions and annotations computed with Prokka and EggNOG mapper, respectively.

### 3.7. Phylogenetic Analysis

To evaluate the evolutionary history of CaHV-1, a maximum likelihood tree based on the general time-reversible model was built (Figure 5) (bootstrap number = 1000, highest log likelihood = −300,835.53). This analysis, performed with MEGA11, involves 23 whole-genome sequences (22 CaHV-1 and 1 FHV-1), and strongly supports the identified relationships among the clades.

In addition, expanding on the CaHV-1 tree structure, snippy-multi was applied in order to compare the Sardinian strains with the other 20 CaHV-1 sequences, collected from the work of Lewin and colleagues. The Italian CaHV-1 genomes resulted in a high sequence homology (>99%). This similarity is evident in the phylogenetic tree, since these sequences share the same tree branch (Figure 6). Moreover, they cluster together with the ELAL-1 and BTU-1 strains, which were collected from US and Brazilian samples, respectively, enlarging the completely separated clade that has been already discussed in the aforementioned article [1].

### 3.8. Variant Analysis

Results of the snippy-multi were also used to perform a variant analysis to evaluate the main differences among the various strains (Figure 7) and simultaneously identify variants characterizing Sardinian specimens.

The Sardinian samples differed from the reference in more than 285 variants, most of them are single nucleotide polymorphisms (SNPs) that are associated with various impacts. Furthermore, 19 of these variants were found exclusively in Sardinian strains, with 15 of them having an impact on coding sequences (Appendix A).

Specifically, five of these variants were classified as synonymous and nine as missense: Thr2646Ile, Asn2731Thr, Asn2739Thr, Asn2747Thr, Asn2755Thr, and Asn2763Thr in pUL36 (large tegument protein); Asp119Asn in pUL25 (DNA packaging tegument); Ala715Thr in pUL22 (envelope glycoprotein H); and Phe19Leu in UL8 (helicase-primase subunit).

Finally, the last mutation is an insertion of two TCC codons (c.273_278dupGGAGGA) in the US2 gene (virion protein US2) corresponding to two glutamate residues in pUS2 (p.Glu91_Glu92dup).

In addition, the tandem repeats present in our strains were compared to the reference strain CaHV-1/0194 genome as reported in Table 2. The results showed that tandem repeats located between 62,849–62,994, 97,625–97,773, and 25,110–124,962 bp of our strains differ in the length of the partial units: from 34 bp, 12 bp, and 12 bp in the reference to 30 bp, 13 bp, and 13 bp in our strains, respectively.

## 4. Discussion and Conclusions

This is the first whole-genome sequencing study of canine herpesvirus in Italy. This work is intended to enrich that of Lewin et al. [1], where a new clade (clade 2) of CaHV-1 strains, constituted by the Brazilian (BTU-1) and the American (ELAL-1) genomes, was detected. In this earlier work, the authors hypothesized that ELAL-1 originated from recombination events involving BTU-1. In agreement with this, we strongly believe that the viral strain presented in this study arrived in Sardinia through direct dog interactions or through semen used in artificial insemination. In fact, natural breeding of French Bulldogs can be difficult because of their unique anatomy and unbalanced body structure. As a matter of fact, it is not surprising that artificial insemination is the most commonly practiced breeding method in French Bulldogs.

Phylogenetic analysis included Sardinian specimens within the clade 2, while variant analysis revealed that our strains shared most of the features characterizing BTU-1 and ELAL-1. These features include a very low GC content (31.6%) when compared to the other Herpesviridae even if shared with all CaHVs, a high number of variants compared to the CaHV-1 reference, and a particular variant enrichment in the 5′ end of the genome, between UL56 and UL47 (top right area of Figure 7). Moreover, we also found the same pattern of non-synonymous variants characterizing ELAL-1 in UL5 (helicase-primase helicase subunit), UL23 (thymidine kinase), UL30 (DNA polymerase catalytic subunit), and UL42 (DNA polymerase processivity subunit) genes. These genes are of great interest since they are involved in various degrees of viral replication and are particularly studied in human herpes simplex virus due to the presence of antiviral resistance-associated missense mutations [39,40].

Therefore, in order to deepen the impact of those variants at the protein level, consensus sequences of animal alpha-herpesviruses, including our strains and feline herpesvirus 1 (FeHV-1), phocid herpesvirus-1 (PhoHV-1), equine herpesvirus-1-4-8-9 (EHV-1-4-8-9), bovine herpesvirus 1-5 (BoHV-1-5), and suid alphaherpesvirus 1 (SuHV-1 or PrV), were compared with those of the human simplex virus (HHSV-1, HHSV-2).

Our results revealed that the Pro299Ser mutation in UL23 does not fall into conserved and catalytically active regions and can be thus classified as a polymorphism not involved in drug resistance (as reported in the study by Labrunie et al., 2019 [39] on HHSV-1 isolates from hematological patients). It was also shown that the Pro299Leu mutation in UL30 falls within the first 361 amino acid residues in a non-functional domain. Interestingly, the Lys447Glu mutation in UL5 reconstitutes the amino acid Glu present in most alpha-herpesvirus species of animal origin, and in HHSV-1 and HHSV-2, and does not appear to be a mutation associated with any particular function nor does it affect drug resistance association. Lastly, the Lys223Arg mutation in UL42 occurring between two amino acids, with the same basic polar charge, does not interfere with the non-specific binding of the UL42 protein to DNA [41,42].

Nevertheless, Sardinian viral specimens constitute a separate strain with specific molecular markers as they are characterized by nineteen unique single nucleotide variants (SNVs) that have not yet been found in other CaHV-1 strains. A total of fifteen out of the nineteen SNVs fall in coding regions: nine are missense variants, five are synonymous variants, and one is an in frame insertion (Appendix A).

In order to define specifically whether these mutations are located in functional protein regions, we compared our amino acid sequences with homologous sequences already studied in the literature. The amino acid variant Thr2646Ile, characteristic of our two strains, affects the large tegument protein of 3176 amino acids encoded by the UL36 gene. According to the study carried out on HHSV-1 by Schipke et al., 2012 [14], the protein variation appears to fall in the C-terminal portion, between the 2430 and the 2893 amino acid residues. These are involved in binding to the capsid protein pUL25 in order to target capsids to cytoplasmatic membranes. In pUL36, we found another distinctive pattern of variants in a repeat region within the coding sequence. This latter consists of five repeat units of 24 bp and a partial unit of 15 bp between the 36,181 bp and 36,315 of the genomes. As shown in Table 2, a mutation in the seventh nucleotide (A > C) of each of the last four units leads to a change in the amino acid sequence from KNLSVPES to KTLSVPES (Asn > Thr).

The alignment of the pUL25 sequence of our strains, with those derived from other animals (genus *Varicellovirus*: PhoV-1, FeHV-1, EHV-1,4,6,8,9, BoHV-1,5) and from human (genus *Simplexvirus*: HHSV1-2) α-herpesviruses, revealed that mutation Asp119Asn falls in an area of different homology between sequences, specifically between residues 111 and 130, which from crystallography studies in HHSV-1 corresponds to a secondary loop structure [15].

Based on research on the pUL22 structure, the missense mutation Ala715Thr detected in our strains is located in the H3 domain, more specifically in a highly conserved C-terminal B-sandwich structure [16]. However, it is also reported [17] that several mutations in this region did not alter the protein function. The comparison of results from mutagenesis studies on the pUL8 protein [18] and from subunit interaction maps [43] suggests that the Phe719Leu mutation, which is located in the C-terminal portion (just upstream of the 33 last amino acids; 717–750 aa), in conjunction with the first 23 amino acids from the N-terminal portion are crucial for viral DNA synthesis.

Finally, a short sequence of only three nucleotides was found to be repeated five times in the coding region of the US2 virion protein. In Sardinian strains, the repetitive TCC nucleotide triplet starts at nucleotide 273 of the CDS and encodes five consecutive glutamic acid residues instead of the three observed in the reference CaHV1 strain.

Variant analysis did not find any polymorphism in the terminal repeats (TRS/TRL) (Figure 7). This is not surprising considering the reads mapping to those regions were discarded, since they were flagged as multi-mapping (MAPQUAL = 0). To overcome this issue, which was already considered and solved in the assembly step, specific parameters should be applied during alignment (i.e., MAPQUAL = 0) and variant calling (i.e., VARQUAL = 0, keeping heterozygous variants) steps, but with the risk of calling uncorrected variants. Nevertheless, the possible variability found within repeats should not add much more information, as phylogenetic analysis performed by multiple sequence alignment of whole-genome sequences (mafft + raxmlHPC) confirmed the results obtained by using a variant-based approach (snippy-multi). In addition, another distinctive feature is also found in the internal inverted repeat (IRL) (nt 97,504–97,541) with a double T deletion in the initial sequence. Anatomo-histopathological exams underline similar patterns of lesions described in the literature [44], consistent with fulminant infection of CaHV-1.

Overall, when comparing our results to previously published studies we found that all unique polymorphisms in Sardinian strains, although affecting coding regions, do not seem to alter protein functionality. For this reason, even if this variant analysis seems not of immediate use, it has allowed for specific markers identification of the Sardinian strains.

With regards to the trend of the disease, it appears that in bitches, antibody titers increase after puberty and are probably protective against a future pregnancy, in case of recent contact. In fact, CaHV-1 is considered a poorly immunogenic virus since antibodies against it disappear a few months after exposure [45,46,47]. This virus becomes latent in neurological tissue, reactivating due to stressful events or strong immunosuppressive drug therapies, with intermittent virus elimination through the respiratory and genital tracts of affected animals [9]. So, seronegative pregnant bitches are at high risk of infection since the virus is enzootic. The strains isolated in our work were both obtained from kidney homogenates of dead puppies. The pathognomonic lesions linked to a high level of mRNA of CaHV-1 determined in several organs are consistent with a fulminant CaVH-1 infection as a consequence of transplacental transmission. Given the serological positivity of almost all of the individuals in the herd, we hypothesize that the infection may have been transmitted to the mother through semen from an infected male at the time of artificial insemination, the most commonly practiced fertilization method in this breed, and transmitted to the puppies at the time of parturition. Another hypothesis involves the transmission from a member of the herd other than the sperm donator, prior to the pregnancy. The latter hypothesis is less likely to be true since it would be difficult to find serological positivity in almost any individual of the herd if the infection was not recent.

In light of the high serological prevalence rates of the virus in several of the surveyed countries [21], it is becoming increasingly clear that surveillance, vaccination, and study of the molecular and phylogenetic characteristics of the virus are highly needed for the prevention of infection and pathology.

In conclusion, clinical and postmortem examination coupled with the increased sensitivity of new molecular diagnostic tools are essential in achieving a differential diagnosis of neonatal pathologies in puppies, and in gaining insights into CaHV-1 transmission in order to improve biosecurity measures in breeding facilities worldwide.

In addition, phylogenetic and variant analyses enabled the complete genome-wide characterization of two novel CaHV-1 strains, thereby increasing knowledge about the recently discovered second clade [1] of this widely circulating, but still understudied virus.

## Figures and Tables

**Figure 1 viruses-16-00209-f001:**
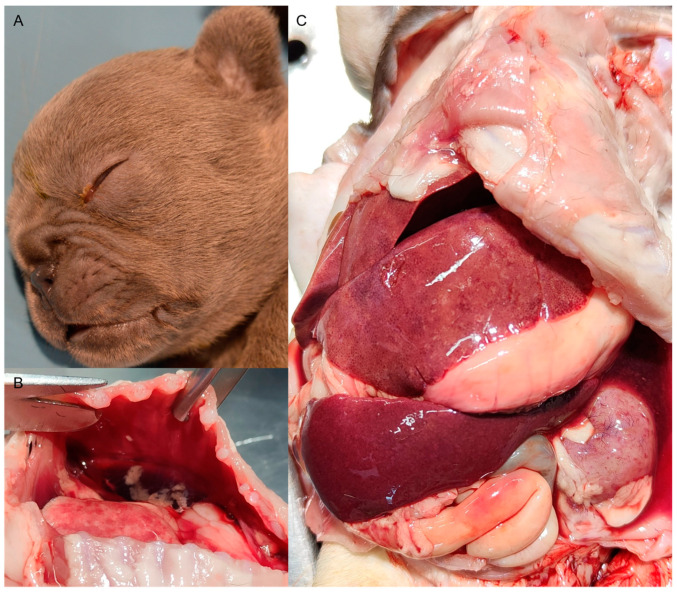
(**A**) Representative external post-mortem examination of a French Bulldog puppy infected with CaHV-1, showing mild serous ocular discharge. (**B**) Representative canine herpesvirus associated macroscopical lesions in an infected puppy. The thoracic cavity is full of abundant sero-hemorrhagic liquid with fibrin clots. The non-collapsed lung shows coalescing pale foci. (**C**) The abdominal cavity shows sero-hemorrhagic ascites, an enlarged liver with multifocal necrosis and hemorrhages, splenomegaly, as well as a congested kidney displaying widespread petechial hemorrhages in the cortex.

**Figure 2 viruses-16-00209-f002:**
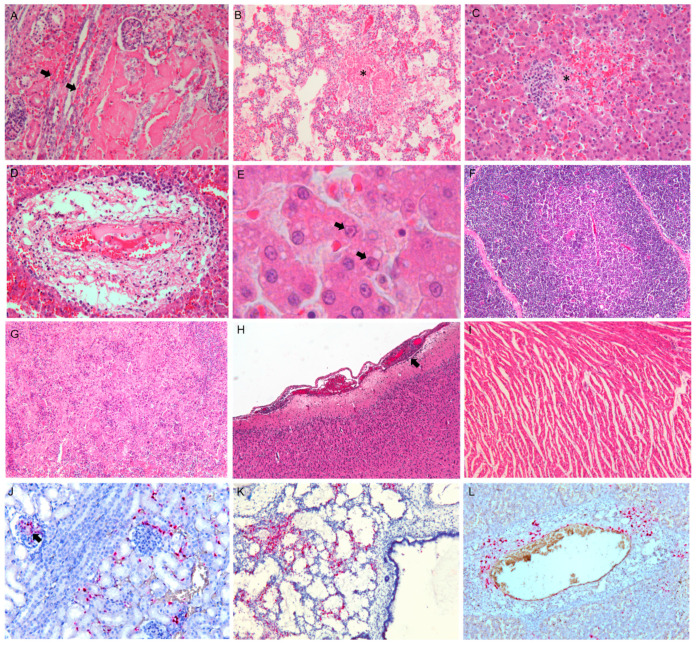
Representative canine herpesvirus-associated microscopical findings in infected puppy. (**A**) Kidney. Degeneration to coagulative necrosis involving tubular epithelium is observed along with hemorrhages in the interstitial space (black arrows). (**B**) Lung. Severe fibrinous hemorrhagic interstitial pneumonia characterized by alveolar multifocal necrosis (asterisk), fibrin deposition, hemorrhages, and mild inflammatory cell infiltration. (**C**–**E**) Liver. (**C**) Multifocal hemorrhagic necrosis (asterisk), lack of cellular details and sparse diffused foci of mononuclear cell infiltration. (**D**) Blood vessel wall is expanded by edema with fibrin necrotic debris. Few lymphocytes, plasma cells, macrophages, and neutrophils are observed. (**E**) Eosinophilic intranuclear inclusions (black arrows) are noted in the hepatocytes adjacent to the necrotic areas. (**F**) Thymus. Atrophy of the medullary zone. (**G**) Spleen. Depletion of white and red pulp with absence of follicles and focal areas of necrosis were observed. (**H**) Brain. Mild congestion and focal non-suppurative meningitis (black arrow) were detected in the cortex. (**I**) Heart. Interstitial edema in the ventricular myocardium. Hematoxylin and eosin. (**J**–**L**) Kidney, lung, and liver. (**J**) CaHV-1 mRNA localization (red) is associated with glomerular cells (black arrow) and infiltrating inflammatory cells. Moreover, CaHV-1 mRNA is associated with (**K**) lung mononuclear inflammatory cells and is found in liver perivascular necrosis areas (**L**). RNA-scope in situ hybridization (ISH).

**Figure 3 viruses-16-00209-f003:**
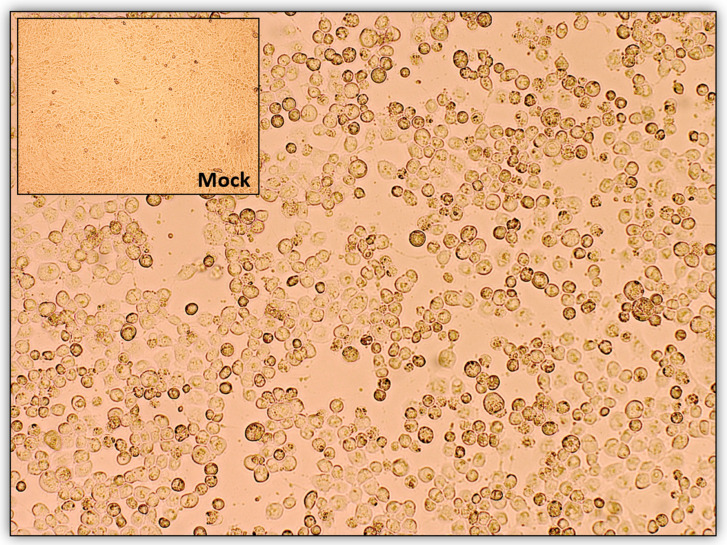
Cytopathic effect of CaHV-1 on MDCK cells consisting of rounding, ballooning and detachment (20× magnification). Mock (non-infected cells).

**Figure 4 viruses-16-00209-f004:**
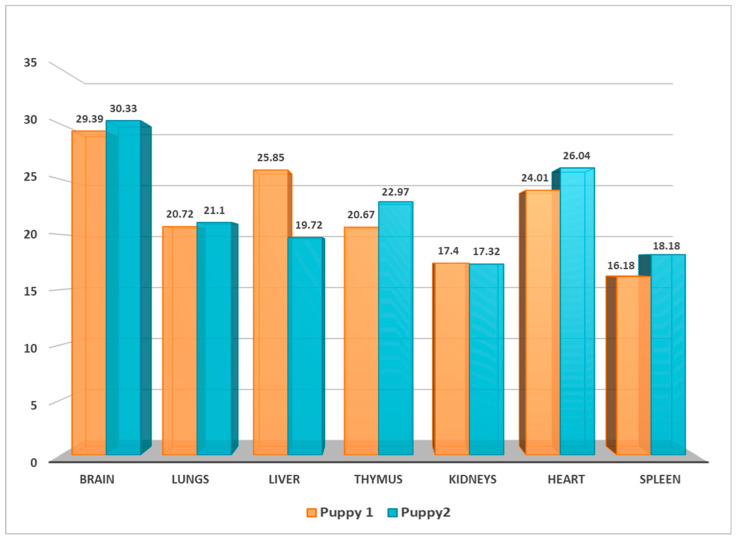
CaHV-1 real-time PCR Ct-value bar graph of puppy organs infected by CaHV-1.

**Figure 5 viruses-16-00209-f005:**
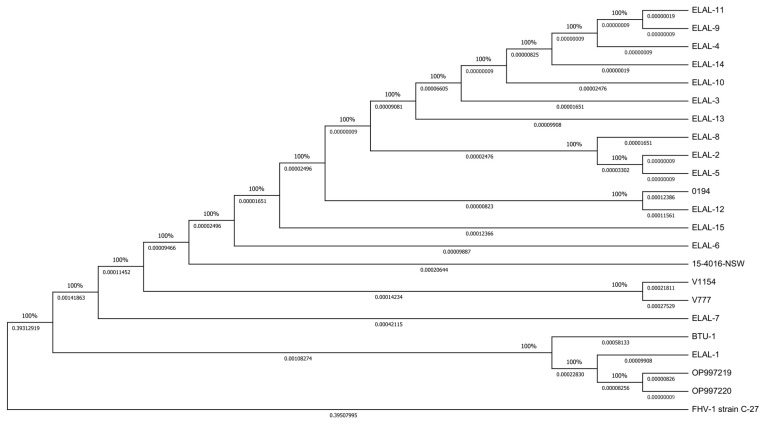
CaHV-1 evolutionary analysis. Maximum likelihood tree of all analyzed CaHV-1 strains with an FHV-1 outgroup (bottom). The percentage of trees in which the associated taxa clustered together is shown above the branches, while the proportion of sites where at least one unambiguous base is present in at least one sequence for each descendent clade is shown below the branches.

**Figure 6 viruses-16-00209-f006:**
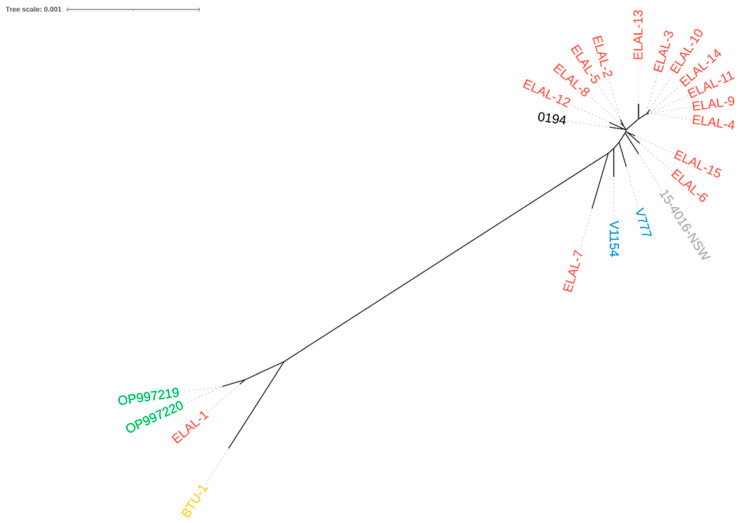
CaHV-1 phylogenetic analysis. Unrooted phylogenetic tree based on whole-genome sequence comparison of 22 CaHV-1 strains. Colored labels represent sample isolation location (red from the US, gold from Brazil, blue from the UK, gray from Australia, and green from Italy). The sample isolation location of the reference (black) is unknown.

**Figure 7 viruses-16-00209-f007:**
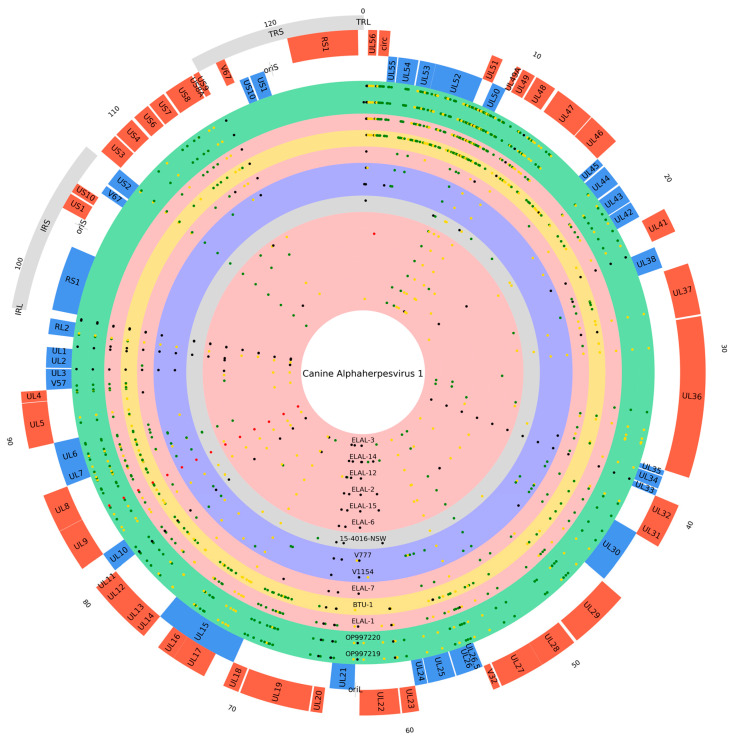
CaHV-1 strain variant analysis. Graphical representation of variant analysis drawn with a customized version of python pyCircos (https://github.com/ponnhide/pyCircos (accessed on 20 September 2022). The three outer rings depict repeat features (light gray), positive strand (tomato), and negative strand (cornflower blue) genes. Inner circles show the distribution of variants (dots) in the analyzed strains along the reference genome. Ring background color indicates the location of specimen isolation: red from USA (ELAL-1, ELAL-2, ELAL-3, ELAL-6, ELAL-7, ELAL-12, ELAL-14, ELAL-15), gold from Brazil (BTU-1), blue from UK (V777, V1154), gray from Australia (15-4016-NSW), and green from Italy (OP997219, OP997220). Dot color indicates the variant impact, assigned according to the work of Cingolani and colleagues [38]: red for high, yellow for moderate, green for low, and black for modifier. Genome size is reported in Kbp in the external area.

**Table 1 viruses-16-00209-t001:** Assembly metrics related to the Geneious assembled genomes.

Feature	OP997219	OP997220
Number of Contigs	1	1
N50	125,214	125,214
L50	1	1
%GC	31.59	31.59
Avg. Coverage Depth	492	1172
Number of N	7	11
Predicted Genes	77	77
Annotated Genes	73	73

**Table 2 viruses-16-00209-t002:** Tandem repeats in CaHV-1 strain 23,612 genomes.

Location (bp)	Unit Sequence (5′–3′)	Length (bp)	Unit (bp)	Unit No.	Partial Unit (bp)
130–227	TCCCATAACCCC	98	12	8	2
16,288–16,489	TATCAACACCCGCGGAGAACAAGCTCCAGAATTCAGTCC ACGGAGTCTGAGTCTAATTTTGAC	202	63	3	13
35,409–35,680	AAACAACCAACCACAGTCCAGCAACCCGCC	272	30	9	2
36,205–36,315	CCAAGACCCTCAGCGTCCCAGAGT	111	24	4	15
36,316–36,622	ACGGGGAACCCGAGGACGCCAGCG	307	24	12	19
62,849–62,994	TACTGGGATCGGGGGGTTGAGGACGCGGATGGTTCACGG CACGCCGGATCGAGCGTGA	146	58	2	30
93,005–93,420	ACTATTAGAATTAACACTCTTACGTCTAGATTGTTTCAACTCTGATGCATCTCCCAACTTCTCTGTAGAATA	407	72	5	47
97,625–97,773	ATAGTCCAACCCCCTTAGGCCCCGCCCACTCAAT	149	34	4	13
125,110–124,962					
102,512–102,679	CATGTTGATCCTCCCTCTTTGTGTATCCCATTTGCGTGTGTAATTAGGCCGC	168	52	3	12
120,223–120,056					
102,688–103,066	ATATTTAAATTGGCTGCCATGTAAACCCTCCCTCTATTACGTGTGTAATTTAC	379	53	7	8
120,047–119,669					
103,035–103,321	CCCTCTATTACGTGTGTAATTTACATATTTAATTGAATCA	287	40	7	7
119,700–119,414					
105,508–105,765	AAATCTATGAATG	257	13	19	10
117,203–116,970					

## Data Availability

The sequences of the CaHV-1 whole genomes obtained during the present study are openly available in the GenBank nucleotide sequence database under the accession number: OP997219, OP997220.

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
