# Peer review of "Whole-Genome Sequencing of Two Canine Herpesvirus 1 (CaHV-1) Isolates and Clinicopathological Outcomes of Infection in French Bulldog Puppies"

_viruses, 2024, doi:10.3390/v16020209_

Round 1

Reviewer 1 Report

Comments and Suggestions for Authors

There is a great deal of interest in those infections that remains partially unknown for years. Canine herpesvirus 1 is one of them, since it is becoming increasingly clear that the virus can cause many more issues than previously expected in particular in breeding facilities.  The manuscript by Rocchigiani and colleagues provides additional information about the genome structure and clinical outcome of this infection in an Italian region.

Although the aim of the work is overall clear and the results and methods are strongly supported and well presented, the text needs some language revision together with a better organization of presented data (in particular for M&M section to make the reading easier). Some minor English suggestions (but not exhaustive) are indicated in the report

Here are some minor suggestions.

In general, please normalize the acronym of Canid herpesvirus 1 (CHV1 or CaHV1) in the text. According to ICTV the CaHV-1 is recommended.

1. Introduction

I would suggest adding a (very) brief part regarding CHV1 diagnosis in the introduction section, since an extensive part of the work is focused on that.

2. M&M

Line 73. In the methods, the authors mentioned the clinical syndrome but never explain that. Please specify which kind of symptoms the authors recorded.

Line 76. Considering also the conclusions reached by the authors, it should be interesting to have some info about the mother of the litter. Age and type of breed (naturally or artificially).

Line 77. Change “dead” with “death”.

Line 84. Remove the list of samples since it is redundant.

Paragraph 2.4 needs to be rephrased in order to make it clearer (in particular from line 121 to 126).

Line 122. Change “e” with “and”.

Line 130 -131. I suggest moving this sentence to line 81.

3. Results

Figure 2 caption should be rephrased and extensively grammar checked.

Line 270. Change “as well the” with “as well as the”.

Figure 4. I would suggest modifying the image with a histogram plot with relative expression/ mean CT values with scale in the y-axis and type of sample in the x-axis

Table 1. I suggest eliminating the table 1 since data can be presented as text and table is not effective.

Line 314-319. Please rephrase. Consider to change “clustered near” with “cluster together with”

Figure 5. The unrooted tree is a good way to depict the phylogenetic relationship without making assumption about the ancestor. However, considering the overall scope of the work, to give a better indication of the directionality of changes I suggest using an outgroup genome, as for example the FHV1, in order to have a better understanding of the phylogenetic reconstruction. Moreover, branch labels with frequency and bootstrap support can be added.

4. Discussion

Line 394 to 400. I agree with authors also considering that French bulldog breeding naturally can be challenging due to their unique anatomy and unbalanced structure. Thus, the artificial insemination is the most commonly practiced method i these breeds.

I would add the statement that support this thesis (also repeated in line 491)

Line 500. Please change “sensibility” with “sensitivity”

Line 501. I would modify this sentence since this is not the case of differential diagnosis presented in the work. No other pathogens were investigated. If yes, please specify in the M&M section (at least in chapter 2.5) otherwise rephrase.

Comments on the Quality of English Language

the text needs some language revision together with a better organization of presented data

Author Response

Dear reviewer, thank you for the valuable suggestions.

First of all, as you suggested ("The text needs some language revision together with a better organization of presented data"), we revised the English text.

Below you can find the answers to your suggestions (changes in the text are colored red ):

Question: In general, please normalize the acronym of Canid herpesvirus 1 (CHV1 or CaHV1) in the text. According to ICTV the CaHV-1 is recommended.

Answer: done

  1. Introduction

Question: I would suggest adding a (very) brief part regarding CHV1 diagnosis in the introduction section, since an extensive part of the work is focused on that.

Answer: added (lines 39-42)

  1. M&M

Question: Line 73. In the methods, the authors mentioned the clinical syndrome but never explain that. Please specify which kind of symptoms the authors recorded.

Answer: added (lines 79-80).

Question: Line 76. Considering also the conclusions reached by the authors, it should be interesting to have some info about the mother of the litter. Age and type of breed (naturally or artificially).

Answer: added (lines 82-84)

Question: Line 77. Change “dead” with “death”.

Answer: done (line 84)

Question:  Line 84. Remove the list of samples since it is redundant.

Answer: done (lines 94-95)

Question:  Paragraph 2.4 needs to be rephrased in order to make it clearer (in particular from line 121 to 126).

Answer: done (lines 131-137)

Question: Line 122. Change “e” with “and”.

Answer: changed

Question: Line 130 -131. I suggest moving this sentence to line 81.

Answer: done (moved to lines 89-91)

  1. Results

Question: Figure 2 caption should be rephrased and extensively grammar checked.

Answer:  Following the request of the reviewer we rephrased the caption FIG 2( lines 266-281)

Question:  Line 270. Change “as well the” with “as well as the”.

Answer: done (line 296)

Question: Figure 4. I would suggest modifying the image with a histogram plot with relative expression/ mean CT values with scale in the y-axis and type of sample in the x-axis

Answer: done.

Question: Table 1. I suggest eliminating the table 1 since data can be presented as text and table is not effective.

Answer: deleted.

Question: Line 314-319. Please rephrase.

Answer: done ( lines 340-349)

Question: Consider to change “clustered near” with “cluster together with”

Answer: done (line 355)

Question: The unrooted tree is a good way to depict the phylogenetic relationship without making assumption about the ancestor. However, considering the overall scope of the work, to give a better indication of the directionality of changes I suggest using an outgroup genome, as for example the FHV1, in order to have a better understanding of the phylogenetic reconstruction. Moreover, branch labels with frequency and bootstrap support can be added.

Answer: new figure added (Figure 5).  Explanation of methods, discussion of the result and two references have been added.

  1. Discussion

Question:  Line 394 to 400. I agree with authors also considering that French bulldog breeding naturally can be challenging due to their unique anatomy and unbalanced structure. Thus, the artificial insemination is the most commonly practiced method in these breeds. I would add the statement that support this thesis (also repeated in line 491)

Answer: done (added in lines 336-339 and 535-536)

Question:  Line 500. Please change “sensibility” with “sensitivity”

Answer: done (line 545)

Question: Line 501. I would modify this sentence since this is not the case of differential diagnosis presented in the work. No other pathogens were investigated. If yes, please specify in the M&M section (at least in chapter 2.5) otherwise rephrase.

Answer: differential diagnosis added at line 150-151, section M&M

Reviewer 2 Report

Comments and Suggestions for Authors

The authors present an interesting study presenting the results of two CHV-1 isolates from Italy. The study builds on existing knowledge regarding the emergence of an interesting subtype of CHV-1. While the underlying methods used appear mostly appropriate, I have some suggestions as to how the manuscript can be improved.

Comments

While the text is mostly well written, there are recurring spelling and grammatical errors which would benefit from the input of a native English speaker. For example, the abstract states ’20 days aged’ where the term ’20 day old’ would be more grammatically appropriate. Similarly, the next sentence states ‘The Italians genomes resulted very similar…’ when a more grammatically appropriate sentence would be ‘The Italian CHV-1 genomes were near identical (>99%).’ These kind of errors continue throughout the text. I have not provided any more comments on the language in the manuscript, but it should be fully reviewed and modified by a native English speaker (ideally one who is familiar with the subject area) prior to resubmission.

Line 109 – MDBK – this acronym is not defined but generally stands for madin darby bovine kidney cells. Why did the authors use a bovine cell line? The authors later use MDCK. Please clarify. Insufficient detail regarding the maintenance of the cell line is provided. Also, it is unclear what is meant by ‘positive controls’ in this context. This section contains numerous spelling and grammatical errors.

Line 129 – the rtPCR referred to here in the citation included a plasmid quantification curve. Did the authors do this? If so, why is Ct and not copy number referred to in the figures? Copy number would be the preferred unit.

Line 182 – de novo assemblies may be less accurate than reference based assembly. Why was this approach chosen? And can the authors provide a comparison analysis to show the reader that the de novo is more appropriate that the reference based assembly in this instance?

Line 268 – While the authors state the tissues were positive for CHV-1, it is unclear how this was determined, particularly since no mention of copy number is included (and little information is provided on controls). Why are there 2 sets of numbers for each bar?

Line 358 – the authors should explicitly state when changes are ‘predicted’ versus confirmed in an experimental setting.

Discussion – While comparing the CHV-1 genome to HSV is an interesting idea, I personally think that this kind of analysis may be a bit of a stretch. HSV-1/2 are very different to animal herpesviruses like FHV-1 and CHV-1. For example, antiviral resistant mutants are fairly common in HSV-1/2 but have never been described in CHV-1 or FHV-1. I would consider reviewing the discussion and keeping only the essential aspects of discussion of this type. Perhaps consider instead comparison of specific variants in the Italian isolates to existing isolates. For example, why do 2 distinct clades exist?

Figure 1 – the discharges referred to in the image are not easily seen. Please consider rewording or improving photograph quaility. Typo – ‘no-collapsed’

Figure 2 – could not make out details referred to in the legend. Please consider increasing size of image, and including marks/arrow/circles on figures to point out the relevant details.

Figure 3 – Consider describing what is shown in image.

Figure 4 – Was a dilution curve used for rt-PCR? If yes, consider using copy number, not Ct value.

Table 1 – Is this sufficiently complicated data to warrant a table?

Figure 5 – Consider adding outgroup to tree. Will help reader understand directionally of evolution and is the accepted norm in such studies.

Figure 6 – Legend does not make it clear which genomes are included in analysis in first sentence. Figure quality in reviewed version makes it hard to see any detail of colors referred to in legend.

Table 3 – contains raw data from assessment software, eg, ‘Effect’ column. Please consider cleaning up data so in format suitable for main text or consider moving to supplemental.

Table 4 – Unite or unit? What does the location refer to? Alignment or sequence?

Comments on the Quality of English Language

While the text is mostly well written, there are recurring spelling and grammatical errors which would benefit from the input of a native English speaker. For example, the abstract states ’20 days aged’ where the term ’20 day old’ would be more grammatically appropriate. Similarly, the next sentence states ‘The Italians genomes resulted very similar…’ when a more grammatically appropriate sentence would be ‘The Italian CHV-1 genomes were near identical (>99%).’ These kind of errors continue throughout the text. I have not provided any more comments on the language in the manuscript, but it should be fully reviewed and modified by a native English speaker (ideally one who is familiar with the subject area) prior to resubmission.

Author Response

Dear reviewer, thank you for the valuable suggestions.

First of all, as you suggested (The text needs some language revision together with a better organization of presented data), we revised the English text.

Below you can finf the answers to your suggestions (changes in the text have the color blue):

Question: Line 109 – MDBK – this acronym is not defined but generally stands for madin darby bovine kidney cells. Why did the authors use a bovine cell line? The authors later use MDCK. Please clarify.

Answer: corrected (line 119)

Question: Insufficient detail regarding the maintenance of the cell line is provided. Also, it is unclear what is meant by ‘positive controls’ in this context. This section contains numerous spelling and grammatical errors.

Answer: rephrased (lines 131-137)

Question: Line 129 – the rtPCR referred to here in the citation included a plasmid quantification curve. Did the authors do this? If so, why is Ct and not copy number referred to in the figures? Copy number would be the preferred unit.

Answer: We used Real-time PCR validated by Decaro et al. only to detect the presence/absence of the viral genome in the various tissues of the two pups, so that the value was expressed in threshold cycles. Performing target viral genome quantification by constructing a reference curve with plasmid standards was not deemed necessary for the purpose of this work.

Question: Line 182 – de novo assemblies may be less accurate than reference based assembly. Why was this approach chosen? And can the authors provide a comparison analysis to show the reader that the de novo is more appropriate that the reference based assembly in this instance?

Answer: lines 321-327

Question: Line 268 – While the authors state the tissues were positive for CHV-1, it is unclear how this was determined, particularly since no mention of copy number is included (and little information is provided on controls).

Answer: The real-time developed and validated by Decaro was taken as a reference for amplification of the CaHV-1 target using only the primers and probe described in the work. The method was then subjected to a secondary validation, modifying the internal positive control (we used VetMAX Xeno Internal Positive Control DNA, certified by Applied Biosystems) and simultaneously amplifying the CaHV-1 target.  The internal positive control (IPC also referred to as IC) during the extraction step (VetMAX Xeno Internal Positive Control DNA (Applied Biosystems) was used to exclude DNA loss or the presence of PCR-inhibiting factors.   To assess the stability of the IC itself, two nontemplate-containing controls (NTC-IPC) were prepared in each amplification plate, the Ct value of the IPC must be ≤37 in the NTC-IPCs, negative controls (KN) and samples. Negative extraction controls (Kn) consisting of Phosphate Buffered Saline pH 7.3±0.1 used in organ preparation for DNA extraction. Positive controls consisting of DNA extracted from strain CHV-1 F205 produced and titrated on Madin-Darby canine kidney (MDCK) cells, kindly provided by the Istituto Zooprofilattico Sperimentale Lazio-Toscana. The amplification of the CaHV-1 target by means of the probe and primers was positive when Ct values are below 40, negative if above 40.

Question: Why are there 2 sets of numbers for each bar?

Answer: Regarding the presence of two Ct/bar values in Fig. 4, they refer to each of the CaHV-1-infected puppies under study.

Question: Line 358 – the authors should explicitly state when changes are ‘predicted’ versus confirmed in an experimental setting.

Answer:  Of the 19 variants observed (all confirmed), 15 were all confirmed to have an impact on the coding sequence.

Discussion

Question: While comparing the CHV-1 genome to HSV is an interesting idea, I personally think that this kind of analysis may be a bit of a stretch. HSV-1/2 are very different to animal herpesviruses like FHV-1 and CHV-1. For example, antiviral resistant mutants are fairly common in HSV-1/2 but have never been described in CHV-1 or FHV-1. I would consider reviewing the discussion and keeping only the essential aspects of discussion of this type. Perhaps consider instead comparison of specific variants in the Italian isolates to existing isolates. For example, why do 2 distinct clades exist?

Answer: In the discussion, given the placement of the two Sardinian strains in the same clade of the phylogenetic tree along with the ELAL-1 and BTU strains (albeit in different branches), we analyzed the mutations that determine their close correlation and those that contribute to their characteristic polymorphism. In fact, the close correlation is due to the large number of mutations, of which the most important are argued from lines 445 to 467. Starting from line 468 we discuss the characteristic mutations of our strains, which are in fact in a different branch albeit within the same clade. We do not discuss the mutations that differentiate our strains from those clustered in the larger clade, because their mutations are different from ours and mostly uninvestigated. It is not our intent to make a characterization of all strains in the phylogenetic tree, but to characterize our strains by distinguishing them from those closely related to them and to evaluate their mutations on the basis of what is in the bibliography, which at present covers only human herpesviruses.Although we knew it was risky, we considered the existing literature on human herpesviruses such as HSSV1/2. Some works discuss missense mutations that have affected the same proteins in our strains and /or those of the same clade, as described by Labrunie et al and Burrel et al, due to drug selective pressure. Although we agree that the human herpes genomes (HHSV-1/2) are very far from those animals, especially CaHV-1 and FeHV-1, with the construction of consensus sequences, we have only made preliminary observations so far, trying to place amino acid substitutions in protein structures where they have occurred. As a result, we tried to build a preliminary polymorphic profile of our strains not only based on genetics and phylogenetic distance from others, but with a protein profile. However, we know that in order to have scientific value our data will need the analysis of many strains derived even from dogs undergoing antibiotic therapy.

Question:  Figure 1 – the discharges referred to in the image are not easily seen. Please consider rewording or improving photograph quaility. Typo – ‘no-collapsed’

Answer: We have increased the size of the FIG  1 and rephrased its caption (lines 243-249)

Question: Figure 2 – could not make out details referred to in the legend. Please consider increasing size of image and including marks/arrow/circles on figures to point out the relevant details.

Answer: We have rewritten the caption of the FIG 2 (lines 266-281)

Question: Figure 3 – Consider describing what is shown in image.

Answer: done (lines 291-292)

Question: Figure 4 – Was a dilution curve used for rt-PCR? If yes, consider using copy number, not Ct value.

Answer: we did not make a dilution curve. You can find a more accurate explanation in the answer about the question on line 268

Question: Table 1 – Is this sufficiently complicated data to warrant a table?

Answer: deleted.

Question: Figure 5 – Consider adding outgroup to tree. Will help reader understand directionally of evolution and is the accepted norm in such studies.

Answer: new figure 5 added.

Question:  Figure 6 – Legend does not make it clear which genomes are included in analysis in first sentence. Figure quality in reviewed version makes it hard to see any detail of colors referred to in legend.

Answer: figure 6 (now 7) reattached and checked the resolution making its understanding clearer.

Question: Table 3 – contains raw data from assessment software, eg, ‘Effect’ column. Please consider cleaning up data so in format suitable for main text or consider moving to supplemental.

Answer: Table 3 moved to Supplemental as Table S1.

Question: Table 4 – Unite or unit? What does the location refer to? Alignment or sequence?

Answer: Unit.

Round 2

Reviewer 1 Report

Comments and Suggestions for Authors

All the points have been fixed

Comments on the Quality of English Language

Minor revisions are needed

Reviewer 2 Report

Comments and Suggestions for Authors

Thank you for providing the revised version of your manuscript for review. It is much improved following revision.

However, the authors have not sufficiently addressed one specific area of concern - the RT PCR used for detection of CHV-1 DNA. The authors state in their reply that no quantification curve using a known standard was used. Therefore, the assay was not performed as was described in the citation, and this must be explicitly stated in the revised manuscript. In addition, using a Ct value cutoff of 40 without such a curve is problematic as false positives are likely to occur 35-40. If this is what the authors considered 'positive', this should be explicitly stated in the manuscript. Did the authors actually have any Ct values over 35? If not, why not consider 35 the cutoff and remove this concern (still needs to be defined in the manuscript)?

Comments on the Quality of English Language

As a minor issue, spelling errors (eg, line 299) and some grammatical errors (eg, line 24, line 25) remain in the manuscript, although this version is improved in this regard. Please note that I have not reviewed the entire manuscript for spelling and grammar and this should be performed (again) prior to resubmission.

Author Response

Question: The authors have not sufficiently addressed one specific area of concern - the RT PCR used for detection of CHV-1 DNA. The authors state in their reply that no quantification curve using a known standard was used. Therefore, the assay was not performed as was described in the citation, and this must be explicitly stated in the revised manuscript. In addition, using a Ct value cutoff of 40 without such a curve is problematic as false positives are likely to occur 35-40. If this is what the authors considered 'positive', this should be explicitly stated in the manuscript. Did the authors actually have any Ct values over 35? If not, why not consider 35 the cutoff and remove this concern (still needs to be defined in the manuscript)?

Answer: explained in lines 148-152.

In our work no standard curve was performed but an Absolute Real Time PCR (see line 148), that exclusively detected the presence/absence of viral DNA through a Ct cut-off value of ≤35. The Real Time program consists of 45 reaction cycles (Decaro et al 2010). Samples with Ct less than 40 were considered positive. However, all samples examined in our work had Ct around 16, so we agree to include a ct cut-off of 35 in the manuscript.

Question: As a minor issue, spelling errors (eg, line 299) and some grammatical errors (eg, line 24, line 25) remain in the manuscript, although this version is improved in this regard. Please note that I have not reviewed the entire manuscript for spelling and grammar and this should be performed (again) prior to resubmission.

Answer: line 302 corrected

Lines 24-25 ( now 22-25) rephrased

The entire manuscript has been rechecked for English grammar

Round 3

Reviewer 2 Report

Comments and Suggestions for Authors

Thank you for responding to my queries and congratulations on such an important and interesting paper. I have no further comments or concerns.

Comments on the Quality of English Language

The English in the paper is good,  but the paper should still be reviewed by the editorial team for accuracy.